# Three-Layer Reconstruction of a Full-Thickness Nasal Alar Defect after Basal-Cell Carcinoma Removal

**DOI:** 10.3390/reports7030075

**Published:** 2024-09-09

**Authors:** Kostadin Gigov, Ivan Ginev, Ivaylo Minev, Petra Kavradzhieva

**Affiliations:** 1Section of Plastic Reconstructive and Aesthetic Surgery and Thermal Trauma, Department of Propedeutics of Surgical Diseases, Faculty of Medicine, “St. George” University Hospital Plovdiv, Medical University of Plovdiv, “Peshtersko shausse Blvd 66”, 4002 Plovdiv, Bulgaria; kostadin.gigov@phd.mu-plovdiv.bg (K.G.); ivan.ginev@phd.mu-plovdiv.bg (I.G.); 2Department of Anaesthesiology Emergency and Intensive Care Medicine, Clinic of Anaesthesiology and Intensive Care, “St. George” University Hospital Plovdiv, Medical University of Plovdiv, 4002 Plovdiv, Bulgaria; ivaylo.minev@mu-plovdiv.bg

**Keywords:** nasal reconstruction, full-thickness nasal defect, forehead flap, skin graft, conchal graft

## Abstract

Restoring the integrity of the external nose presents a complex surgical challenge due to its three-dimensional structure and subunit divisions. The most frequent causes of nasal defects include basal or squamous cell carcinoma, animal bites and trauma. The reconstruction approach varies depending on factors such as the defect’s size, the affected subunit of the nose, the condition and quality of the surrounding tissues and the surgeon’s expertise. Commonly employed surgical techniques for nasal reconstruction include the forehead flap, nasolabial flap, Rieger flap, bilobed flap, and free autologous skin graft. We present a clinical case of a patient with a full-thickness nasal alar defect who underwent three-layer reconstruction with a combination of an inverted skin flap- for the internal lining, a conchal graft to substitute the missing alar cartilage and a forehead flap for external coverage. One of the challenges of nasal reconstruction surgery is to restore a full-thickness defect, especially the internal nasal lining. We offer a different perspective on this problem, showcasing considerable advantages, as there are limited literature data on this method.

## 1. Introduction

Nasal reconstruction is a complex surgical intervention due to the nose’s three-dimensional structure and subunit division. The most common causes of defects are basal cell or squamous cell carcinoma, animal bites, traumas and burns [1,2].

The subunit concept, described by Burget and Menick in 1985, serves as a pivotal point in the restoration of nasal integrity, and states that the external nose comprises the following subunits: dorsum, lateral nasal sidewalls, nasal tip, columella, paired alae and paired soft triangles (Figure 1) [3,4,5].

Depending on the size of the defect, which subunit of the nose is affected, the condition and quality of surrounding tissues and surgical skills, there are various ways to reconstruct the nose. Frequently used surgical techniques include the forehead flap, nasolabial flap, Rieger flap, bilobed flap, composite conchal graft or skin graft [1,2]. Limitations to current techniques include patient comorbidities, the involvement of more than one subunit, a size defect and prolonged surgical treatment, especially when performing a three-staged forehead flap [4,5,6].

This case report represents a patient with a full-thickness nasal alar defect after removing basal cell carcinoma. The surgical approach included a combination of internal lining restoration with an inverted skin flap, the replacement of the absent alar cartilage with a conchal graft and external coverage with a forehead flap, executed in two stages.

A systematic review from 2021, analysing 176 articles, discusses diverse approaches to restoration of nasal defects according to the subunit principle, with the forehead flap being predominantly employed for deformity size >1.5 cm [6]. However, to the authors’ knowledge, there are no substantial data regarding the utilisation of an inverted skin flap, conchal cartilage and forehead flap in combination for a two-staged, three-layer reconstruction approach to full-thickness nasal alar defects. A significant problem in nasal reconstruction is the creation of a reliable internal lining, which could support a cartilage graft and prevent valve collapse. Therefore, this case report provides an insightful perspective into the existing knowledge of nasal reconstructive methodology.

## 2. Case Presentation

An 80-year-old male with a full-thickness nasal alar defect was admitted for reconstructive surgery. He had previously undergone Mohs surgery in the department of Otorhinolaryngology due to Morpheus-type basal cell carcinoma, with clear resection margins and no radiotherapy. He had a history of previous BCC removal and reconstructive surgery in the temporal region (Figure 2). After discussing the possible surgical options, the method of choice was a three-layer defect reconstruction. The intervention started with creating an inverted skin flap from the adjacent sidewall subunit to reconstruct the inner layer toward the nasal cavity (Figure 3). Conchal cartilage from the ipsilateral ear, accessed through a retro-auricular incision, was utilised to reconstruct the second layer and restore missing alar cartilage. The graft was shaped to achieve the desired contour (Figure 4a).

A forehead flap vascularised by the supratrochlear artery was employed for the outer layer reconstruction. Using a template gauze, the edges of the defect were marked to assess flap size accurately. The flap was approximated to the wound defect without tension using simple interrupted sutures. The donor site was left for secondary intention healing with a tie-over dressing (Figure 4b,c).

Postoperatively, there were no complications, and the vascular perfusion of the flap remained intact. The tie-over dressing was removed on the seventh day, revealing healthy granulation tissue filling the secondary defect. On the 20th day of the follow-up, the intervention was considered 100% successful, with the robust vascularisation of the forehead flap and complete granulation tissue coverage at the donor site (Figure 5a,b). A new tumour lesion near the donor site of the flap was noticed and considered for removal at the next stage of treatment.

At the next stage of the treatment, after a 5-week interval from the first surgery, we performed the flap pedicle division and the excision of the new advanced lesion on the forehead, which was sent for frozen section analysis. The pathologist reported Basal Cell Carcinoma with clear resection margins. The remaining defect was covered with a split-thickness skin graft from the inguinal area. The skin graft was secured with an interrupted suture and a tie-over gauze. The donor site was closed primary. On the seventh day, the stitches and the dressings were removed, and we evaluated the seamless integration of the forehead flap within the nasal subunits after pedicle division, with excellent vascularisation and 100% success of the skin graft (Figure 6a,b). The fixed histological sample confirmed the result from Mohs’ procedure with clear resection margins and radical surgical treatment. The patient underwent no additional radiotherapy.

## 3. Discussion

There are different surgical methods for nasal reconstruction, depending on the defect size and subunit involvement. Most of them, however, lack systematic literature reviews [6]. Previously published works do not provide explicit data on the usage of inverted skin flap from adjacent sidewall subunit, conchal cartilage and forehead flap all together.

In this discussion, the authors provide a short anatomical review of the nose, examine the existing surgical techniques and showcase the advantages and limitations of the method in question.

The nasal ala is composed of three anatomically separate layers, including the skin- as an external layer; the middle layer, containing the alar cartilage; and the internal layer, which comprises the mucosal lining [2]. Restoring each layer is crucial to ensure the external nasal valve functionality and obtain an aesthetic result, defined by a smooth transition between the different subunits of the nose and an absence of scar deformity [3]. Combining strategies and procedures is necessary to achieve a satisfying result when reconstructing a three-layer nasal defect. The concept of subunits is critical for the intervention. The anatomical structure of the superior two-thirds of the nasal framework differs from that of the inferior one-third. The cephalic two-thirds represent the nasal dorsum, comprising the sidewall and dorsal subunits, where the skin has different properties than the skin in the caudal one-third [4]. A high priority is placed on maintaining the distinguishable borders of the subunits. Therefore, the defect’s shape and size can be altered to precisely match the aesthetic concept, providing that the resultant scars are aligned with the edges of these subunits. An exception of the subunit rule is when there is a mid-line vertical scar of the proximal two-thirds of the nose, which divides the dorsum into two halves. Nevertheless, this scar does not cause any deformity and is rarely visible to the untrained eye. The most commonly affected nasal subunits are the alae, counting for approximately one third of the nasal defects [5,6]. The size of the defect is one of the deciding factors for the reconstruction method. It is also important to acknowledge the skin quality, as the skin of the distal one-third has insufficient mobility and, therefore, only small-size defects, less than 0.5 cm in diameter, are indicated for primary closure [7]. The forehead flap is often recognised as the most effective method for treating large composite defects of the nose, especially when combined with the use of cartilage grafts [8]. Nasal reconstruction with a forehead flap is a common surgical method performed in elderly adults, producing favourable outcomes with almost indiscernible scarring. To achieve success, it is essential to precisely assess the wound defect and length of the flap to approximate it with no tension. Insufficient flap length and tension pose a great risk for comprised vascularisation [9]. A correctly performed surgery can yield a long-lasting outcome, with inconspicuous scarring, restored nasal valve function and satisfactory skin colour and texture match. The forehead flap reconstruction is typically executed in three stages, including the flap thinning in the second stage and pedicle division in the third stage. This time investment is a notable drawback, as it is associated with an increased anaesthesia risk and patient discomfort and morbidity. The flap pedicle base is designed to lie on the middle portion of the eyebrow, including the supratrochlear artery and branches of the angular artery. The elevation of the flap commences at the subcutaneous plane, where the skin paddle is directly dissected from the frontalis muscle and galea aponeurotica rather than in the subgaleal plane. Elevation in the subgaleal plane should be considered in patients with metabolic and peripheral vascular diseases. After flap elevation, the dissection continues obliquely toward the medial supraorbital ridge. The periosteum is included at the base of the flap pedicle, enhancing the length and reducing flap tension. Pedicle division is performed at a minimum of a 3-week interval, ensuring the necessary period for the flap vascularisation from the wound bedside [9,10,11].

The nasolabial flap is a great alternative to the forehead flap with regard to alar defects. It is safe, has an excellent vascular supply, has a good skin colour and texture match and is relatively easy to execute [4,6]. However, the nasolabial flap might have been insufficient in our case, so the forehead flap was our method of choice for external coverage.

In the nasal alar reconstruction of a full-thickness defect, it is essential to replace the absent alar cartilage and ensure nasal valve patency, preventing the ala’s collapse. The intermediate layer is restored with cartilage, which can be harvested from either a rib or the ear, depending on the size and shape of the defect [12]. The conchal cartilage is a suitable donor tissue to replace the alar cartilage, as it has elastic properties, moldability and similarity to the anatomical arch of the nasal ala [13]. Another great advantage of this method is that the whole base of the concha can be removed without causing auricular deformity [14]. Free composite conchal grafting is another reconstruction method, enabling a one-stage surgical procedure with good aesthetic outcome and valve patency. However, the unpredictable survival of the graft is a significant limitation. Another drawback is that the size of the defect should not exceed 1.5 cm in its largest diameter [15].

If there is a full-thickness nasal alar defect, the first step during surgery is restoring the absent endonasal mucosa. This step plays an important role, as the inner layer provides a supportive base for the cartilage graft and prevents its contracture and deformity, as well as the collapse of the external valve. The internal layer reconstruction can be achieved by using the nasolabial folded flap, forehead folded flap, septal mucoperichondrial flap, cutaneous hinge flaps, turbinate flap, full-thickness skin grafts, etc. [4,15,16,17,18].

We consider a great advantage in our approach the ability to use the inverted skin flap from the nasal sidewall without causing facial disfigurement, as the forehead flap can cover the alar defect and the secondary sidewall defect. Another benefit is avoiding the use of a septal mucoperichondrial flap, which could possibly cause damage to the internal nasal structures and septum [17]. Also, the inverted flap can be easily and quickly performed. Together, the inverted flap and the forehead flap create stability and hermeticity for the conchal graft with sufficient coverage.

A limitation of our reconstructive method is the necessity of a second stage after several weeks to perform the pedicle division, which prolongs the overall treatment.

## 4. Conclusions

The patient’s follow-up revealed a long-lasting result with restored anatomy, defined nasal subunits with smooth transitioning, normal breathing function without external valve collapse and satisfactory appearance.

To the best of our knowledge, there are limited data on the type of approach we used, which involved the employment of the aforementioned techniques in combination. We believe it can be successfully applied in similar cases with full-thickness nasal alar defects and provide an opportunity to address all absent layers in the first stage.

Universal techniques and algorithms for nasal alar reconstruction do not exist, and patients should be assessed individually in order to choose the most suitable treatment method.

## Figures and Tables

**Figure 1 reports-07-00075-f001:**
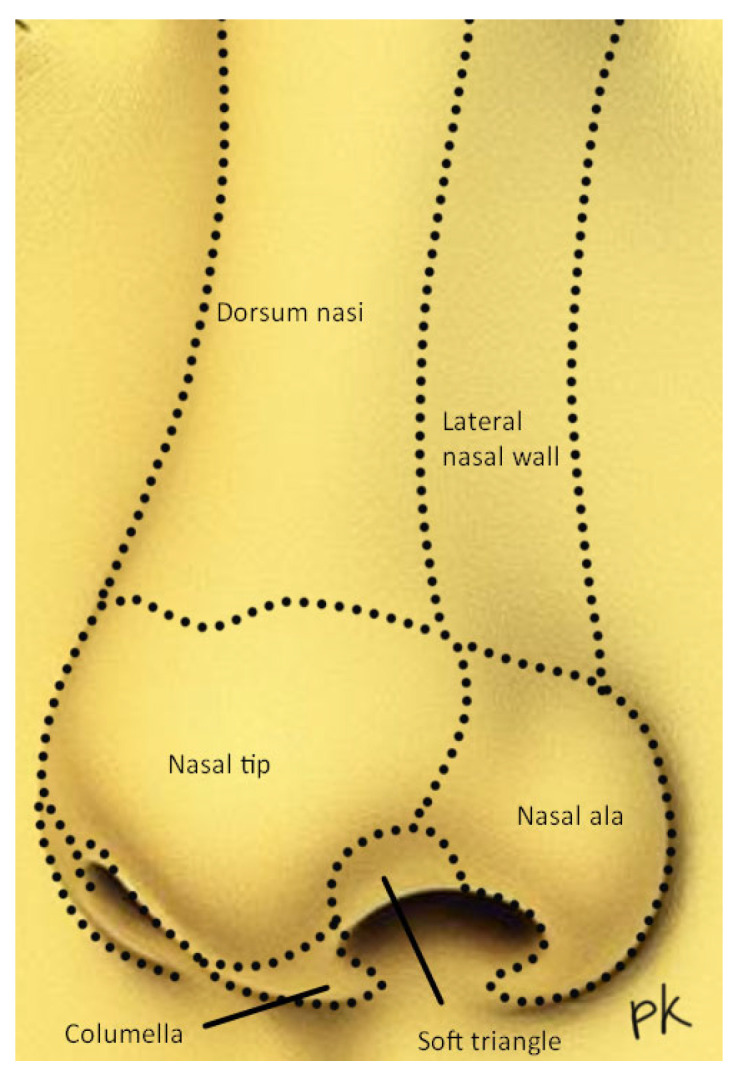
Aesthetic subunits of the nose.

**Figure 2 reports-07-00075-f002:**
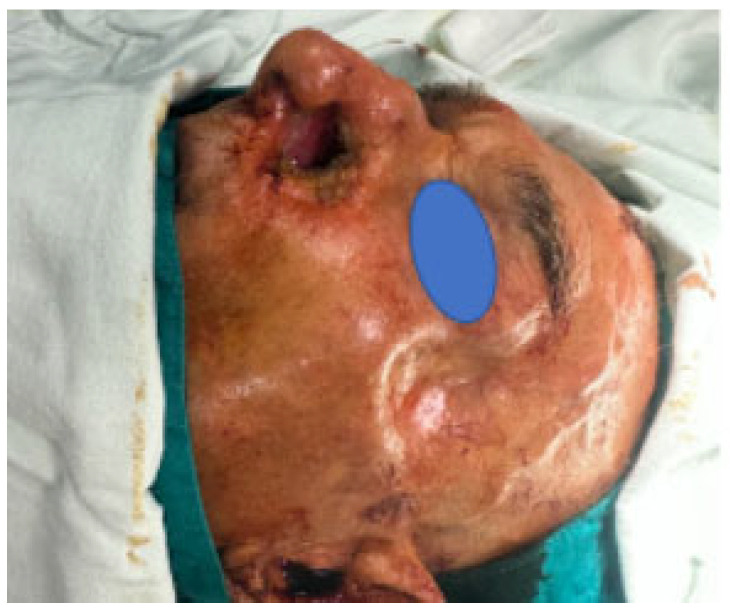
Full-thickness nasal alar defect.

**Figure 3 reports-07-00075-f003:**
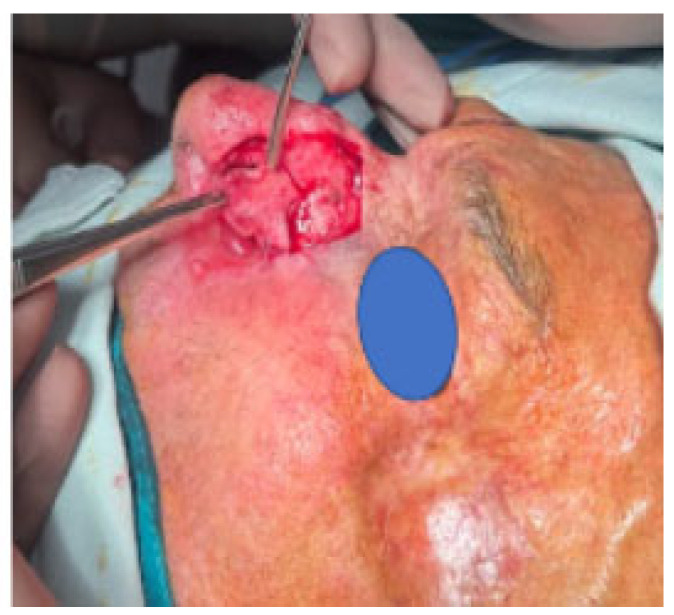
Inverted skin flap.

**Figure 4 reports-07-00075-f004:**
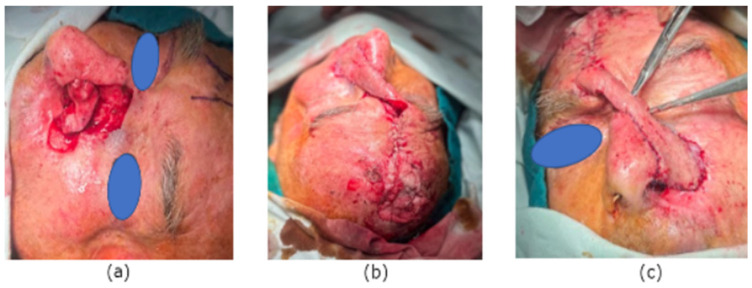
(**a**) Conchal cartilage. (**b**) Tie-over dressing for donor site. (**c**) Forehead flap for the external layer.

**Figure 5 reports-07-00075-f005:**
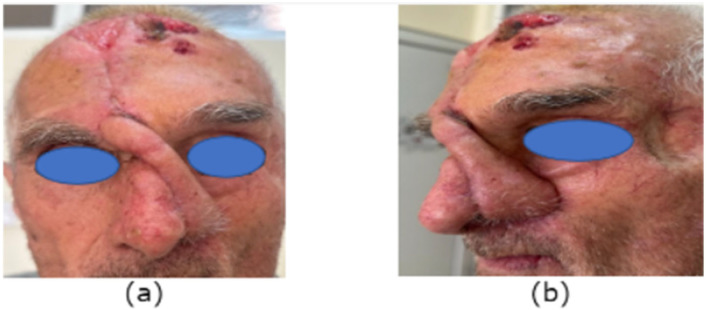
(**a**) Result on the 20th day. (**b**) Result on the 20th day.

**Figure 6 reports-07-00075-f006:**
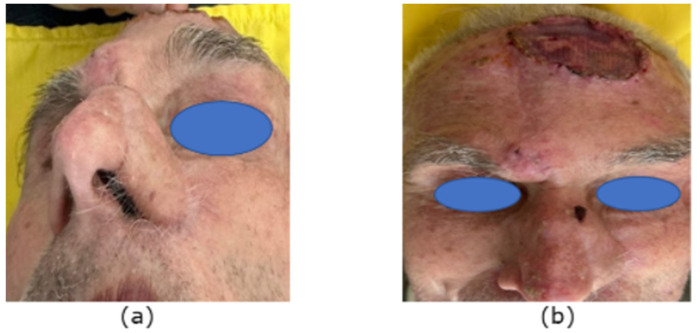
(**a**) Forehead flap after pedicle division—7th day. (**b**) Pedicle division and skin graft—7th day.

## Data Availability

The original data presented in the study are included in the article, further inquiries can be directed to the corresponding author.

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
