# Peer review of "Three-Layer Reconstruction of a Full-Thickness Nasal Alar Defect after Basal-Cell Carcinoma Removal"

_reports, 2024, doi:10.3390/reports7030075_

Round 1

Reviewer 1 Report

Comments and Suggestions for Authors

1. The abstract looks incomplete. A suggestion to revise with some key findings and novelty/concluding sentence.

2. The Introduction section must be revised and rewritten, stating the related literature, research gap, and novelty of the reported work.

3. A suggestion to merge back-to-back images for better representation (E.g., Figs. 1, 2 and Figs. 3, 4, 5).

4. The discussion section is good, but a suggestion is to add some aligned literature studies to showcase the significance of this work.

5. The conclusions section has more general statements than the overall crisp of the reported work. A suggestion is to revise the section effectively.

6. Information missing for author's contributions, funding, institutional review board statement, informed consent statement, and data availability statement.

Comments on the Quality of English Language

Need some changes and checks during the production phase. 

Author Response

Dear Reviewer,

Thank you for your insightful comments and remarks! We have made the necessary corrections, and we are adding our response below.

Comment 1: "1. The abstract looks incomplete. A suggestion to revise with some key findings and novelty/concluding sentence." We agree to the statement and we have added the missing information.

Comment 2: "The Introduction section must be revised and rewritten, stating the related literature, research gap, and novelty of the reported work."

Introduction section was thoroughly revised and rewritten, adding the missing elements.

Comment 3: A suggestion to merge back-to-back images for better representation (E.g., Figs. 1, 2 and Figs. 3, 4, 5). 

We agree with the statement, as the images did not look completely structured. The necessary merging was done.

Comment 4: The discussion section is good, but a suggestion is to add some aligned literature studies to showcase the significance of this work. 

  • We have added the necessary literature and made some updates to the references in order to be fully aligned with current knowledge.

Comment 5: The conclusions section has more general statements than the overall crisp of the reported work. A suggestion is to revise the section effectively.

  • This section was also revised and some specific statements and conlcusions were mentioned.

Comment 6: Information missing for author's contributions, funding, institutional review board statement, informed consent statement, and data availability statement.

Missing information was filled in the case report template.

*All the changes are highlighted in the updated manuscript template.

Reviewer 2 Report

Comments and Suggestions for Authors

Please, verify what is new in your case presentation.

Was the forehead flap appropiate as adonor site?. There was local tumour in area of donor site of flap.

Why nasolabial flap was not your choice?. It is more safe, can provide 3 layers covrage, accepted donor site and could be used for this patient as a single stage.

Author Response

Dear Reviewer! 

Thank you for the interesting comments, as they provide food for thought! Below, we present our responses to each comment.

Comment 1: "Please, verify what is new in your case presentation."

To the best of our knowledge, literature lacks data about this type of three-layer approach with attention to inverted skin flap for internal lining. We believe it is a unique perspective, as it provides alar stability and bedsite for alar cartilage. Also, compared to septal mucoperichondrial flaps,  it spares other mucosal tissue. We have added the necessary literature to support our statements.

Comment 2: Was the forehead flap appropiate as adonor site?. There was local tumour in area of donor site of flap.

We consider the forehead flap appropriate, as the donor site was not affected by the local tumor. Also, the forehead flap provides significant opportunities to be designed and moulded to achieve the necessary nasal contour, according to the subunit principle.

Why nasolabial flap was not your choice?. It is more safe, can provide 3 layers covrage, accepted donor site and could be used for this patient as a single stage.

The forehead flap is proven to be safe throughout the decades and there is great experience with it. We have chosen this method as it is able to provide more tissue coverage and there is great opportinuty to design the flap in accordance to subunit principle. Indeed, the nasolabial flap has the aforementioned advantages, however, this method might be insufficient in our particular case and if we use it we will cause trauma to the midface, as well. In elderly people, it is proven that scars from forehead flap heal very well, whereas utilizing the nasolabial flap could potentially cause asymmetry between the nasolabial folds. Also, as we have made a forehead defect after lesion excision in the 2nd stage, and if we had used the  nasolabial flap, in case an infection arises in both sites, the situation would have been more difficult to handle.

Thank you for the remarks and I hope we have been explicit!

All the necessary chnages in the report have been highlighted.

Reviewer 3 Report

Comments and Suggestions for Authors

Dear authors, I read your valuable article carefully and it seems that some explanations may help readers. 

1. Introduction: please give some information about the limitations of current techniques for ala reconstruction.

How much was the time interval between the first and second surgery?

3. please write about the limitations of your reconstructive approach 

Comments on the Quality of English Language

It is good.

Author Response

Dear Reviewer,

Thank you for your comments and remarks! Below, we present our response to your suggestions.

 Comment 1. Introduction: please give some information about the limitations of current techniques for ala reconstruction.

We have revised and rewritten, according to the comment, and provided the necessary background and limitations.

How much was the time interval between the first and second surgery?  The division took place after 5 weeks.

3. please write about the limitations of your reconstructive approach 

Limitation to our approach is the involvement of a second stage, which prolongs the treatment.

All the necessary changes of the report have been highlighted.

Reviewer 4 Report

Comments and Suggestions for Authors

It gives me great pleasure to have the opportunity to review this interesting case report. The authors have presented a case of full-thickness defect of the nasal alar after BCC removal, which was successfully treated with three-layer reconstruction. The paper is well written and intriguing. I have a few comments and questions below.

1.           In the case report and technique, "ENT Department" should be written as a full word without abbreviation.

2.           What do you think about other flaps such as subcutaneous pedicle island flap using nasolabial fold for external layer reconstruction?

3.           In the discussion, "cephalic" and "caudal" or "superior" and "inferior" should be used instead of "proximal" and "distal" to describe nasal anatomy. In addition, a figure should be included to aid understanding of your explanation of nasal anatomy.

Author Response

Dear Reviewer, 

Thank you for your kind comments and remarks! Below, we are writing our responses.

"It gives me great pleasure to have the opportunity to review this interesting case report. The authors have presented a case of full-thickness defect of the nasal alar after BCC removal, which was successfully treated with three-layer reconstruction. The paper is well written and intriguing. I have a few comments and questions below."

  1. In the case report and technique, "ENT Department" should be written as a full word without abbreviation.""  - The necessary correction was made.
  2. What do you think about other flaps such as subcutaneous pedicle island flap using nasolabial fold for external layer reconstruction? - Very interesting question. We discussed this suggestion in our team. This is indeed, one of the best approaches to alar defects. The flap provides excellent vascularity, similar skin color and texture, and has high rate of success. In our case, we decided to use the forehead flap, as we needed more sufficient amount of tissue to perform this method and cover the cartilage. With the nasolabial flap we might have not been able to achieve this.
  3. In the discussion, "cephalic" and "caudal" or "superior" and "inferior" should be used instead of "proximal" and "distal" to describe nasal anatomy. In addition, a figure should be included to aid understanding of your explanation of nasal anatomy. - The necessary corrections were made.

All changes in the case report were highlighted.

Round 2

Reviewer 1 Report

Comments and Suggestions for Authors

The authors have incorporated all the changes and addressed the queries in the revised manuscript. I recommend the acceptance of the revised manuscript for publication.

Comments on the Quality of English Language

Minor editing can be done by the production team.

Reviewer 3 Report

Comments and Suggestions for Authors

Dear authors

thanks a lot for revision

Reviewer 4 Report

Comments and Suggestions for Authors

The authors responded to all reviewer’s comments and questions properly. I don’t have any other comments.